# APOLLO: A Self-Guided Multi-Agent System for Scientific Article Generation inspired by Human Thinking

## Abstract

Automatic generation of Wikipedia-like articles through Retrieval-Augmented Generation (RAG) has recently gained increasing attention. While recent advances in Large Language Models (LLMs) show considerable promise for synthesizing complex information, current RAG-based systems suffer from two fundamental limitations: they often rely on shallow retrieval strategies, leading to redundant content, and they lack effective mechanisms for factual verification and content organization. To address these challenges, we present APOLLO, a multi-agent framework specifically designed to generate high-quality, comprehensive articles with citations to the given sources. APOLLO simulates the iterative research and editorial process of human contributors through a set of specialized agents that collaboratively retrieve, fact-check, and structure information. To evaluate our method, we introduce SciWiki-2k, a dataset comprising 2,000 high-quality Wikipedia articles spanning 20 scientific domains. Compared to baseline methods, APOLLO produces articles with significantly improved structural coherence, content diversity, and factual accuracy. Human evaluations further establish the practical value of our approach for generating trustworthy, comprehensive articles. In this work, we target scientific Wikipedia-style articles, using "Wikipedia-like" solely to denote an encyclopedic structure with hierarchical sections and in-line citations rather than coverage of all Wikipedia domains.

**Code** – https://github.com/frosty-compiler/apollo

## 1 Introduction

Large Language Models (LLMs) have demonstrated remarkable capabilities in text generation. However, producing comprehensive, well-structured, and factually accurate articles remains a significant challenge Yang et al. (2023); Liang et al. (2023). Current approaches struggle to maintain coherence across extended content, synthesizing diverse information sources, and ensuring factual grounding throughout the generation process. While Retrieval Augmented Generation (RAG) has emerged as a promising solution to enhance LLM capabilities with external knowledge Lewis et al. (2020); Gao et al. (2024), most existing systems rely on static retrieval (orange dotted line Figure 1). Although recent variants like oRAG Shao et al. (2024) have attempted to improve upon this approach, these systems still lack reflective mechanisms which are fundamental in human research when exploring and synthesizing information.

Recent work has begun to address these limitations by introducing more structured and agent-based frameworks. For instance, STORM Shao et al. (2024) and OmniThink Xi et al. (2025) employ multi-agent frameworks to collect information from diverse perspectives or to simulate tree-based mind maps. These methods enhance topic coverage by collecting information from multiple perspectives or simulating reflective exploration (blue dotted line Figure 1). However, while these methods enhance coverage through dynamic retrieval, they struggle to represent relationships among the retrieved information and to organize it coherently Han et al. (2025). This is important because building a coherent view of a topic requires not only collecting isolated facts, but also understanding how the different concepts relate to each other Booth et al. (2003).

In this work, we use Wikipedia articles as a widely familiar reference point to denote an encyclopedic writing style—hierarchical sections and in-line citations—for scientific topics. APOLLO is

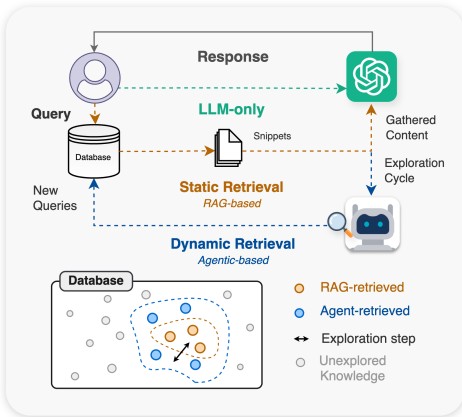

Figure 1: **Overview of retrieval strategies**. Static retrieval systems uses a single query to retrieve relevant content. In dynamic retrieval, gathered information is further analysed to issue new queries and discover new content.

explicitly scoped to scientific subjects that require dense factual grounding and hierarchical organization; we do not claim applicability to non-scientific domains.

Furthermore, human research is inherently iterative and reflective, often involving repeated cycles of exploration, synthesis, and re-evaluation Doyle (1994); Kuhlthau (2004). For instance, when investigating the topic of Ensemble Learning (EL), a researcher might start with a broad overview of the concept and, as their understanding deepens, formulate more focused queries such as "bagging in ensemble methods" or "applications of ensemble learning". This evolving inquiry gradually builds what cognitive scientists refer to as "knowledge structure"; a mental framework in which new information is systematically integrated with prior knowledge Ausubel (1968). Such structures help researchers uncover conceptual relationships and identify information gaps that shape future searches Novak (1998); Chi et al. (2014).

Motivated by these observations, we introduce APOLLO, a multi-agent framework designed to emulate the iterative and reflective patterns of human research and structured writing. APOLLO begins by gathering information about a given topic through iterative proactive retrieval, organizing the retrieved evidence into a Knowledge Graph (KG) that captures entities, relationships, and topical hierarchies. This KG serves as both a record of discoveries and a scaffold for further investigation. Specialized agents analyze the evolving KG to identify missing links and underexplored subtopics, then generate targeted search queries to fill these gaps. This process repeats iteratively, with each cycle enriching the KG with additional relevant information.

After constructing the comprehensive KG, APOLLO transitions to article generation by extracting a hierarchical outline that represents the main concepts and their relationships. For each section, relevant content is retrieved from the information gathered during the knowledge curation step. A specialized writer agent synthesizes this material into an organized, well-supported text, while a reviewer agent systematically examines each draft section, verifies claims against referenced sources, and provides actionable feedback for refinement. This review-and-revision loop continues until all sections are complete and properly cited, mirroring best practices in collaborative academic writing Viégas et al. (2007).

To evaluate the effectiveness of our method, we introduce SciWiki-2k, a comprehensive benchmark for Scientific Wikipedia-style Article Generation (SAG). We evaluate APOLLO across multiple aspects, including knowledge curation, outline generation, and article generation, using automatic metrics, LLM-based qualitative assessments, and human evaluation. Inspired by the fact-checking literature Pang et al. (2023); Min et al. (2023); Thorne et al. (2018), we introduce two novel metrics to measure hallucination and content coverage. We conduct extensive experiments and compare APOLLO with state-of-the-art (SOTA) baselines. The results demonstrate substantial improvements across multiple key evaluation metrics Specifically, APOLLO increases information diversity by 9.2% over OmniThink, achieves a 7.1-point higher coverage rate than STORM on SciWiki-100, and reduces hallucination rate by 18% compared to the next-best baseline. Finally, our human eval-

Table 1: Capability matrix for automated scientific article generation. Criteria: Dynamic Retrieval (issuing new queries based on intermediate analysis), Structured Memory (explicit persistent knowledge representation, e.g., KG), Reflective Thinking (gap detection and planning), and Fact Verification (claim–evidence checking with citations). Symbols denote presence: ✓ = present, ↔ (or ◇) = partial, and → = absent.

| Framework | oRAG | STORM | OmniThink | APOLLO |
|---|---|---|---|---|
| Dynamic Retrieval | ✗ | ✓ | ✓ | ✓ |
| Structured Memory | ✗ | ✓ | ✓ | ✓ |
| Reflective Thinking | ✗ | ✗ | ✓ | ✓ |
| Fact Verification | ✗ | ✗ | ✗ | ✓ |
| **Total Capabilities** | **0/4** | **2/4** | **3/4** | **4/4** |

Table 2: Quantitative analysis of the content found in the SciWiki-2k benchmark dataset.

| Attribute | Value |
|---|---|
| Total Articles | 2000 |
| Scientific Domains | 20 |
| Avg. Number of Sections | 7.8 |
| Avg. Number of All-level Headings | 19.9 |
| Avg. Length of a Section (words) | 483.3 |
| Avg. Length of Article (words) | 3672.7 |
| Avg. Number of References | 71.5 |

uation study confirms that APOLLO outperforms competitive baselines in both overall article quality and factual accuracy, further validating its effectiveness in generating high-quality scientific content.

The main contributions of this work are as follows.

- We present APOLLO, a multi-agent framework that automates long-form, structured article generation through iterative KG construction, reflective gap detection, and agent-based fact verification
- We release the SciWiki-2k, a large benchmark dataset designed for assessing article generation models, and introduce two novel evaluation metrics called Hallucination and Coverage Rate to assess the factuality of the generated text.
- We provide extensive experiments and a human evaluation study to demonstrate that APOLLO outperforms existing baselines in terms of coverage, diversity, and factual reliability metrics.

## 2 PRELIMINARY

### 2.1 PROBLEM DEFINITION

We define the task of SAG as follows. Given a topic $T$, representing a scientific concept (e.g., "Ensemble Learning"), the goal is to produce a comprehensive, factually grounded article $\mathcal{A}$ that explains the concept, outlines its key components, and organizes relevant subtopics and relationships in a coherent structure. We break the task of SAG through a three-stage process: (i) Knowledge Curation, retrieving and organizing relevant information $\mathcal{I} = \text{Retrieve}(T, \mathcal{C})$ from information sources $\mathcal{C}$, (ii) Outline Generation, constructing a structured outline $\mathcal{O} = \text{Construct}(\mathcal{I}, T)$ based on the retrieved information, and (iii) Article Generation, synthesizing the final article $\mathcal{A} = \text{Write}(\mathcal{O}, \mathcal{I})$ using the outline and retrieved information. The challenge lies in ensuring comprehensive coverage, factual accuracy, and coherent organization while avoiding redundancy and hallucinations.

As shown in Table 1, none of the existing methods can fully support this task.

### 2.2 SCIWIKI-2K BENCHMARK

To address the lack of comprehensive benchmarks for SAG, we introduce SciWiki-2k, a curated dataset of 2,000 high-quality Wikipedia articles spanning 20 scientific domains. Unlike existing benchmarks Shao et al. (2024); Jiang et al. (2024c); Liu et al. (2018); Fan & Gardent (2022) that focus on general topics, SciWiki-2k specifically targets scientific concepts, providing high-quality Wikipedia articles as ground truth references for evaluating how well multi-agent systems can generate comparable scientific content.

**Alignment and quality control.** The construction of our dataset follows a rigorous process. We begin by selecting a diverse set of topics representing key trends and core concepts from a broad range of scientific domains. For each topic (T), we map it to a single canonical Wikipedia page using title redirects and disambiguation resolution. We then apply a quality filtering using the ORES

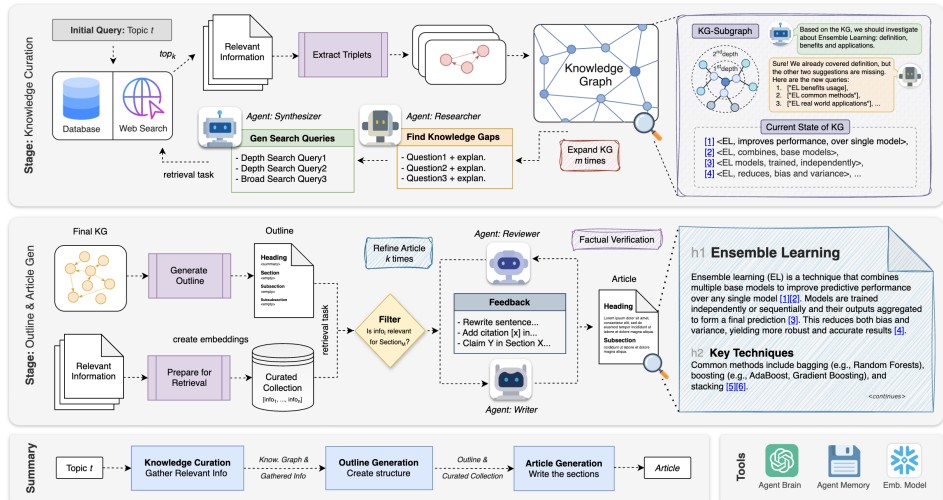

Figure 2: **Overview of APOLLO 's multi-agent framework for SAG**. The pipeline consists of three stages: (i) Knowledge Curation, where information is iteratively gathered and structured as a KG; (ii) Outline Generation, which derives a hierarchical article structure from the graph; and (iii) Article Generation, where writer and reviewer agents collaborate to produce and fact-check each section, resulting in comprehensive, well-cited articles.

API[1], retaining only articles rated as "B-Class", "Good Article", or "Featured Article" according to the Wikipedia grading scheme Wikipedia contributors (2025), thereby excluding low-quality, ambiguous, or insufficiently developed pages. All pages were retrieved between February and March 2025.

After quality filtering, we extract only the main text and section headings from these articles, omitting non-textual elements to standardize the dataset for text-based evaluation. A subsequent manual review is done to ensure the content of each article closely matches the intended scientific topic and domain; articles with misaligned or overly broad coverage are removed.

For reproducibility, we release the full pipeline used to extract these pages, alongside SciWiki-2k[2]. Table 2 shows the composition of our dataset.

## 3 METHODOLOGY

We present APOLLO, a multi-agent framework that automates the generation of comprehensive, factually grounded articles. APOLLO 's pipeline consists of three main stages: (i) knowledge curation, (ii) outline generation, and (iii) article synthesis. Figure 2 provides an overview of our proposed framework.

### 3.1 KNOWLEDGE CURATION

APOLLO begins article generation by proactively gathering and organizing relevant information through an iterative process that constructs KGs to identify coverage gaps and guide targeted exploration. This approach addresses the limitation of simple retrieval strategies that miss valuable related information discoverable through more exploratory search processes Marchionini (2006); Savolainen (2018).

**Initialization Stage.** Given topic $T$, APOLLO supports two retrieval modes: (i) domain-constrained search from a curated corpus $\mathcal{C}_D$ and (ii) open-domain web search. For domain-constrained retrieval, we perform retrieval from the domain-specific collection as follows.

$$\mathcal{I}_0 = \text{Retrieve}(T, \mathcal{C}_D), \tag{1}$$

---

[1]https://www.mediawiki.org/wiki/ORES
[2]https://huggingface.co/SciWiki

where $\mathcal{I}_0 = \{s_{i_1}, s_{i_2}, \ldots, s_{i_k}\} \subseteq \mathcal{C}_D$ represents the top-$k$ most relevant snippets based on cosine similarity. Each snippet $s_i$ contains raw text, embedding vectors, and metadata including source URLs. For web search, we perform an analogous retrieval using search engines.

**Knowledge Graph Construction.** For each retrieved snippet $s_i \in \mathcal{I}_0$, we apply an extraction operator as follows.

$$\Phi : s_i \longmapsto \big\{ (h, r, t) \mid h, t \in \mathcal{E}, r \in \mathcal{R} \big\}, \tag{2}$$

where an LLM ( A.1) extracts a set of triplets in the form of $(h, r, t)$ comprising of a head entity ($h$), a relation label ($r$), and a tail entity ($t$). The extracted triplets define a snippet-level sub-graph $G_i = (V_i, E_i)$ where $V_i \subseteq \mathcal{E}$ represents entities found in $s_i$ and $E_i = \{(h, r, t) \mid h, t \in V_i, \ r \in \mathcal{R}\}$ contains edges linking those entities.

**Graph Aggregation and Normalization.** We construct the initial KG named $\mathcal{G}_0$ by aggregating all sub-graphs ( A.2), as shown in the following.

$$\mathcal{G}_0 = \bigcup_{i=1}^{k} G_i = \left( \bigcup_{i=1}^{k} V_i, \bigcup_{i=1}^{k} E_i \right), \tag{3}$$

Since aggregation may result in duplicate entities, we apply a normalization function $\eta$ as follows.

$$\mathcal{G}_0^* = \eta(\mathcal{G}_0) \tag{4}$$

where $\eta$ is an LLM-based normalizer ( A.3) [3] that merges semantically equivalent entities and their associated edges.

**Expansion Stage.**

The APOLLO framework employs two collaborative agents that proactively expand the knowledge representation in multiple iterations through targeted information gathering, emulating human research patterns Pirolli & Card (1999).

- **Agent 1: Research Question Generator.** The first agent analyses the current KG, $\mathcal{G}_m^*$ at iteration $m$ and produces focused research questions $\mathbb{Q}_m$ as follows.

$$\mathbb{Q}_m = \Psi\big(\mathcal{G}_m^*, \mathcal{M}_Q\big) = \big\{ (q_j, \rho_j) \big\}_{j=1}^{n}, \tag{5}$$

where $\Psi$ denotes the system prompt ( A.4.1) which guides the agent to analyse $\mathcal{G}_m^*$, identify underexplored entities or relations, and generate $n = 10$ research questions including $n_d = 5$ "in-depth" questions targeting specific concepts and $n_b = 5$ "breadth" questions that branch into adjacent areas, (see agent interaction, top right Figure 2). Each question $q_j$ includes a rationale $\rho_j$ justifying its importance. To avoid repetition, the agent maintains a memory set $\mathcal{M}_Q$ that tracks all previously generated questions.

- **Agent 2: Query Synthesizer.** The second agent synthesizes focused search queries from the research questions:

$$\mathbb{L}_m = \Lambda\big(\mathbb{Q}_m, \mathcal{M}_L\big) = \{\ell_1, \ell_2, \ldots, \ell_t\}, \tag{6}$$

where $\Lambda$ is the system prompt ( A.4.2) guiding the agent to (i) decompose each question into salient entities and relations, (ii) paraphrase them into concrete search queries, and (iii) filter out terms already present in its memory $\mathcal{M}_L$. We set $t \leq 10$ to balance the exploration between depth and breadth.

**Retrieval & Graph Update.**

At each iteration, newly generated queries retrieve additional snippets (Eq. 1). These are used to create sub-graphs (Eq. 2), which are merged into the main KG (Eq. 3) after filtering previously seen content (Eq. 4). The agents update their memory sets to track queries and research questions already explored. The expansion continues for $m$ iterations until a maximum depth is reached. The resulting curated collection is:

$$\mathcal{K} = \bigcup_{j=0}^{m} \mathcal{I}_j, \tag{7}$$

which contains all the collected snippets during the iterative process. This curated collection forms the foundation for constructing the final article.

---

[3]Detailed specifications of all LLM-based functions and prompts are available in our open-source repository.

## 3.2 OUTLINE GENERATION

Given the final KG $\mathcal{G}_m^*$ and topic $T$, the framework generates a hierarchical article outline using an LLM-based function called $\Omega$ (Appendix B.1, B.2). This function analyses the structure of the KG and produces a set of headings and subheadings.

$$\mathcal{O} = \Omega(\mathcal{G}_m^*, T) = \{h_1, h_2, \ldots, h_p\}, \tag{8}$$

where each $h_i$ is a main section or subsection, reflecting the entities and relations captured in $\mathcal{G}_m^*$. We instantiate ($\Omega$) with (i) LLM canonicalization of entity surface forms, (ii) embedding-based similarity for alias detection, and (iii) alias/redirect heuristics. This step ensures that the outline matches the conceptual organization found during the knowledge curation step.

## 3.3 ARTICLE GENERATION

In this phase, the outline $\mathcal{O}$ is expanded into the final article $\mathcal{A}$ by gathering section-specific content and refining it through multiple revision cycles.

**Section-Specific Retrieval**

To support each section $h_i$ in the outline, we first retrieve a set of candidate snippets from the collection $\mathcal{K}$ as follows.

$$\mathcal{R}_i = \text{Retrieve}(h_i, \mathcal{K}), \tag{9}$$

where $\mathcal{R}_i$ denotes the top-$k$ snippets most similar to the section heading $h_i$. To ensure relevant information was gathered, we utilize an LLM-based filter C.2 and select a subset of the more relevant snippets as follows.

$$\mathcal{S}_i = \{s \in \mathcal{R}_i \mid \Theta(s, h_i) = \text{relevant}\}, \tag{10}$$

where $\Theta$ determines whether snippet $s$ provides valuable information for writing section $h_i$. The resulting set $\mathcal{S}_i$ serves as the supporting material for generating the content of section $h_i$.

**Iterative Content Generation**   After collecting the relevant information for a section, two agents collaborate to produce the content. For each section $h_i$, let $a_i^{(r)}$ denote the content generated in revision $r$.

- **Agent 3: Writer Agent.** The writer first generates a draft for section $h_i$ using the supporting content found earlier:

$$a_i^{(0)} = \Gamma(h_i, \mathcal{S}_i), \tag{11}$$

where $\Gamma$ is the system prompt that instructs the agent to transform $\mathcal{S}_i$ into a well-organized and factual text which includes in-line citations to the given snippets. After the initial draft, the writer updates the section iteratively based on the feedback of the reviewer agent as follows.

$$a_i^{(r+1)} = \Gamma^{\text{revise}}(a_i^{(r)}, \mathbb{F}_i^{(r)}, \mathcal{S}_i), \tag{12}$$

where $\mathbb{F}_i^{(r)}$ contains a list of bullet points that the writer agent follows to refine the content of the section at revision $r$. An example of this process is shown in Figure 2 (e.g., *Rewrite sentence...*).

- **Agent 4: Reviewer Agent.** The reviewer evaluates the generated content $a_i^{(r)}$ and maintains feedback memory $\mathcal{M}_F$:

$$\mathbb{F}_i^{(r)} = \pi(a_i^{(r)}, \mathcal{S}_i, \mathcal{M}_F), \tag{13}$$

where $\pi$ is the system prompt C.3 that instructs the agent to (i) assess whether cited snippets support claims, (ii) identify inconsistencies, and (iii) produce a structured feedback list $\mathbb{F}_i^{(r)} = \{f_1, f_2, \ldots, f_q\}$ with actionable revision items for the writer agent.

**Reviewer Loop Termination**   The writer–reviewer cycle terminates early when the reviewer returns an empty feedback list (all issues resolved); otherwise it stops at $r_{\max} = 3$. This provides partial adaptivity across sections of varying difficulty. **Article Assembly.**   The final article is constructed by combining all refined sections while preserving the hierarchical structure from the outline. This produces a comprehensive article $\mathcal{A}$ where all claims are supported by evidence from the curated collection $\mathcal{K}$.

Table 3: **Outline & Article Quality Evaluation** Comparison across lexical and LLM-as-judge metrics between Apollo and baseline methods. $\dagger$ denotes significant differences ($p < 0.05$) compared to the best baseline.

| Method | Outline Quality Metrics | | | | |
| --- | --- | --- | --- | --- | --- |
| | Soft Recall | Entity Recall | Content Guidance | Hierarchical Clarity | Logical Coherence |
| *Scientific Corpus* | | | | | |
| oRAG | 86.42 | 37.44 | 3.22 | 3.97 | 3.86 |
| STORM | 87.63 | 37.10 | 3.39 | 3.95 | 3.87 |
| OmniThink | 88.31 | 37.74 | 4.03 | 3.99 | 3.98 |
| APOLLO | 91.82$\dagger$ | 38.52 | 4.16$\dagger$ | 4.00 | 4.01$\dagger$ |
| w/o Reflection | 80.75 | 36.14 | 3.36 | 3.93 | 3.82 |
| *Web Search* | | | | | |
| oRAG | 87.18 | 38.30 | 4.12 | 4.05 | 4.27 |
| STORM | 87.89 | 38.47 | 4.37 | 4.08 | 4.36 |
| OmniThink | 88.45 | 38.41 | 4.44 | 4.04 | 4.51 |
| APOLLO | 92.25$\dagger$ | 42.44$\dagger$ | 4.63$\dagger$ | 4.10 | 4.65$\dagger$ |
| w/o Reflection | 88.92 | 40.92 | 4.25 | 4.02 | 4.33 |

| Method | Article Quality Metrics | | | | | | |
| --- | --- | --- | --- | --- | --- | --- | --- |
| | ROUGE R1 | ROUGE RL | Entity Recall | Interest Level | Coherence Organization | Relevance Focus | Depth Exploration |
| *Scientific Corpus* | | | | | | | |
| oRAG | 41.84 | 14.03 | 5.92 | 2.34 | 4.32 | 3.92 | 3.88 |
| STORM | 42.11 | 14.44 | 6.51 | 1.61 | 4.85 | 4.10 | 4.54 |
| OmniThink | 41.76 | 13.94 | 5.53 | 1.37 | 4.28 | 4.12 | 4.27 |
| APOLLO | 52.10$\dagger$ | 15.81$\dagger$ | 9.17$\dagger$ | 3.29$\dagger$ | 4.92$\dagger$ | 4.90$\dagger$ | 4.94$\dagger$ |
| w/o Filter | 49.17 | 15.51 | 7.35 | 1.99 | 4.74 | 4.57 | 4.77 |
| *Web Search* | | | | | | | |
| oRAG | 39.95 | 13.65 | 5.07 | 2.22 | 4.57 | 3.88 | 4.05 |
| STORM | 42.32 | 14.60 | 5.64 | 2.27 | 4.69 | 4.11 | 4.35 |
| OmniThink | 32.07 | 12.58 | 3.57 | 1.85 | 4.68 | 3.45 | 3.61 |
| APOLLO | 52.01$\dagger$ | 16.88$\dagger$ | 9.44$\dagger$ | 2.92$\dagger$ | 4.84$\dagger$ | 4.87$\dagger$ | 4.46$\dagger$ |
| w/o Filter | 50.28 | 15.92 | 8.21 | 2.02 | 4.83 | 4.79 | 3.89 |

## 4 EXPERIMENTS

**Baselines.** We compare articles generated by our method with those generated by three other baselines, including oRAG Shao et al. (2024), STORM Shao et al. (2024), and OmniThink Xi et al. (2025). oRAG is a two-stage RAG baseline that generates an outline first, then processes each section independently using section-specific retrieval. STORM is a multi-agent system that simulates conversations between perspective-guided agents to gather diverse information before generating articles. OmniThink leverages a hierarchical tree representation to organize and synthesize information for article generation.

**Hyper-parameters.** We implement all the agents using Chain-of-Thought (CoT) Wei et al. (2022) and Zero-Shot (ZS) prompting the `gpt-4o-mini-2024-02-15` model. For reproducibility, we set the *temperature* to 1.0 and *top-p* to 0.9. For web retrieval, we use Brave's API[4] with each query returning up to 3 web pages. For retrieval using the scientific corpus, we use Qdrant[5] with Snowflake embeddings[6]. During the knowledge curation step, we set the maximum depth of exploration to $m = 3$. We set $r_{\max} = 3$ with early stopping when the reviewer's feedback list is empty; this reduces unnecessary revision cycles on easier sections.

All experiments are conducted on a single AWS g5.2xlarge instance (24GiB GPU, 8 vCPUs). To ensure a fair comparison, we allow a maximum of 135 search queries for all baselines. For STORM, we use default values and set the limit of perspectives to 3; for Omnithink, we set the depth of the tree expansion to 3. To ensure robust evaluation, we conduct five independent runs for APOLLO and the baselines under two retrieval settings, including (i) web search using the Brave API and (ii) domain-constrained search using a curated scientific corpus. To build this dataset we use a set of review articles published across 2,700 journals, as well as the content of 43,000 books published in different science domains. For segmenting books and articles into passages, we considered each (sub-)section as a passage[7].

**Dataset.** Following prior work Shao et al. (2024); Jiang et al. (2024a), we evaluate APOLLO using SciWiki-100, a subset of SciWiki-2k dataset constructed by randomly selecting 5 topics from each of 20 scientific domains in this dataset. We generate articles using APOLLO and each baseline for the topics SciWiki-100 dataset. To evaluate whether the extracted KGs effectively capture the information from retrieved snippets, we employ the Measure of Information in Nodes and Edges (MINE) benchmark[8], a dataset designed to evaluate the completeness and factual consistency of KGs extracted from scientific text Mo et al. (2025).

---

[4] https://brave.com/search/api/

[5] https://qdrant.tech/

[6] https://huggingface.co/Snowflake/snowflake-arctic-embed-m-v2

[7] The details of the corpus will be added upon publication.

[8] https://github.com/stair-lab/kg-gen

### 4.1 EVALUATION SETUP

In the following, we explain the metrics used for evaluating the performance of each stage of our framework.

**Knowledge Curation Quality.** We assess the effectiveness of our knowledge curation module by measuring the number of unique sources retrieved and measure information diversity defined by Jiang et al. (2024b) as: $\text{Div}(\mathcal{I}) = 1 - \frac{1}{n(n-1)} \sum_{i \neq j} \cos(\mathbf{e}_i, \mathbf{e}_j)$.

**Outline Quality.** We compare generated section headings against SciWiki-100 reference articles using soft recall and entity recall (named entity overlap via FLAIR NER Akbik et al. (2019)). LLM-as-judge assessments are done using M-Prometheus-7B Kim et al. (2024) to evaluate *Content Guidance*, *Hierarchical Clarity*, and *Logical Coherence* on a 5-point scale. Moreover, we perform an AB preference test comparing APOLLO's generated outlines against the best baseline (i.e., Omnithink) judged by three LLM evaluators (Claude-3.7-Sonnet Anthropic (2024), Llama-3.3-70B-Instruct Touvron et al. (2024), GPT-4o-mini)[9].

**Article Quality.** We assess the quality of the generated content for each section by using Recall, ROUGE-1, and ROUGE-L metrics, considering the content of articles from the SciWiki-100 dataset as gold data. Additionally, we conduct LLM-as-judge assessments on four different metrics, namely *Interest*, *Organization*, *Relevance*, and *Depth*. See Appendix D (Cost Estimation and Average Execution Time tables) for per-article, per-stage cost and time breakdowns across methods (RAG, oRAG, STORM, APOLLO) and backbones (GPT-4o-mini, Claude-3.7), along with measurement assumptions and scalability levers.

**Citation Quality.** We introduce two novel automatic metrics for evaluating citation quality in generated scientific articles: hallucination rate and coverage. Hallucination rate $(1 - \frac{|C_v|}{|C|})$ quantifies the proportion of claims not supported by any evidence linked through in-line citations Min et al. (2023), and coverage $(\frac{|S_v|}{|S|})$ measures the proportion of article sections with at least one claim verifiably grounded in cited retrieved snippets Samarinas et al. (2025). LLM-based entailment is used for automated claim verification. The claim extraction and verification prompts/templates are provided in Appendix C.2.

**Human Evaluation.** We select one topic at random from each of the 20 scientific domains and generate articles using the scientific corpus for both APOLLO and the best baseline (STORM) according to article generation metrics. This results in a total of 40 articles which is scored by Subject Matter Experts (SMEs) using the same 5-point rubric used by our LLM-as-judge for both outline and article quality. SMEs are scientific domain experts with advanced training relevant to the evaluation topics.

**Backbones.** We report KG construction quality with multiple LLMs, including the open-source Llama-3.3-70B and proprietary models (Table 4); full-pipeline outline/article results with Llama-3.3-70B are deferred to camera-ready.

## 5 RESULTS

Starting with the knowledge curation phase we analyse whether our constructed KGs can capture meaningful entities and relationships to further guide the research stage of our framework. To this end, we measure the quality of our KGs construction using the MINE Score. We report the performance of APOLLO using different LLM backbones and two KG construction baseline methods, including KG-Gen Mo et al. (2025) and LightRAG Guo et al. (2024), in Table 4. The results show that our knowledge construction agent outperforms the baselines when using Llama-3.3-70B and GPT-4o-mini LLMs.

Building on these results, we measure how much unique retrieved information APOLLO can discover in the knowledge curation phase. As shown in Table 5, our proposed method consistently retrieves more unique URLs and achieves greater information diversity across both scientific corpus and web search settings. In particular, APOLLO outperforms the next-best method (i.e., OmniThink) by a wide margin in information diversity, confirming the effectiveness of our proactive retrieval agents.

Following our evaluation setup, we assess how well APOLLO constructs article outlines and organizes retrieved information into coherent sections. Looking at the left side of Table 3, we observe

---

[9]We access this models via Amazon Bedrock: claude-3-7-sonnet-20250219-v1 and llama3-3-70b-instruct-v1

Table 4: Results of KG construction quality of our proposed model and baseline model using different LLM backbones.

| LLM Backbone | Method | MINE Score | |
|---|---|---|---|
| | | Normalized | Non-Normalized |
| Claude-3.7-Sonnet | Ours | 0.714 | 0.701 |
| | KG-Gen | **0.725** | 0.680 |
| | LightRAG | 0.709 | **0.705** |
| Llama-3.3-70B | Ours | **0.620** | **0.610** |
| | KG-Gen | 0.580 | 0.550 |
| | LightRAG | 0.535 | 0.542 |
| GPT-4o-mini | Ours | **0.501** | **0.486** |
| | KG-Gen | 0.392 | 0.388 |
| | LightRAG | 0.432 | 0.428 |

Table 5: Average number of unique URLs retrieved by each method.

| Feature | APOLLO | OmniThink | STORM | oRAG |
|---|---|---|---|---|
| **Scientific Corpus** | | | | |
| Num Unique URLs ↑ | 105.71 | 83.27 | 60.12 | 45.45 |
| Info Diversity (%) ↑ | 60.81 | 54.74 | 42.23 | 33.02 |
| **Web Search** | | | | |
| Num Unique URLs ↑ | 88.60 | 63.22 | 59.82 | 19.49 |
| Info Diversity (%) ↑ | 66.02 | 61.64 | 45.13 | 34.92 |

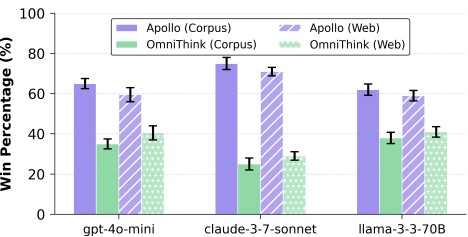

Figure 3: Win rate results from A/B preference tests comparing APOLLO's generated outlines against the best performing baseline across different LLMs evaluators. Claude-3-7-Sonnet displays the highest preference for APOLLO (79.5%). Error bars show standard deviation across 5 runs.

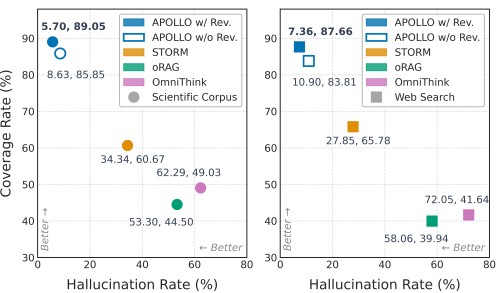

Figure 4: Scatter plot showing the trade-off between coverage rate and hallucination rate for APOLLO and baseline models.

that outlines generated by APOLLO outperform baseline methods using both retrieval settings over all metrics. Higher value of Entity recall and Soft recall for APOLLO compared to the baselines, shows that the outlines generated by APOLLO are more similar to the outlines of the gold standard SciWiki-100 dataset. Moreover, considering the three metrics judged by LLM, we observe a significant improvement in *Content* and *Coherence* over the best baseline method (i.e., OmniThink). Additionally, we use an AB Preference test to validate the superior performance of APOLLO in generating outlines using SOTA LLMs and report the results in Figure 3. As can be seen, using different LLMs, APOLLO constantly wins the OmniThink baseline.

Based on the results presented in Table 3, we can see that APOLLO outperforms the baselines in terms of article-level evaluation metrics. Notably, we observe a significant improvement in *Depth*, *Relevant*, and *Interest* metrics compared to the best performing baseline (i.e., STORM). Moreover, the results of the ablation study show that disabling the reflecting agent and our LLM-based filter function decreases the quality of our generated outlines and articles, respectively. The results of factual accuracy evaluation from Figure 4 demonstrate the importance of our review-and-revision iterative process to avoid the generation of hallucinated content and increase coverage rate. Particularly, we observe that when disabling the Reviewer agent, the articles generated by our method tend to include more unsupported claims and show a marked increase in hallucination rates.

Finally, based on the results of the human evaluation study depicted in Figure 5, we observe that in 7 out of 8 metrics APOLLO is consistently rated higher than STORM (best baseline for article creation). This result further validates the reliability of our automated evaluation metrics.

## 6 RELATED WORK

**Retrieval-Augmented Generation.** RAG enhances LLMs with external knowledge to improve factuality and relevance Karpukhin et al. (2020); Guu et al. (2020). Early work focused on static retrieval pipelines for tasks such as QA Izacard & Grave (2021), summarization Menick et al. (2022), and citation generation Ram et al. (2023). Recent studies explore dynamic retrieval strategies that trigger queries adaptively during generation Jiang et al. (2023); Yao et al. (2023a). However,

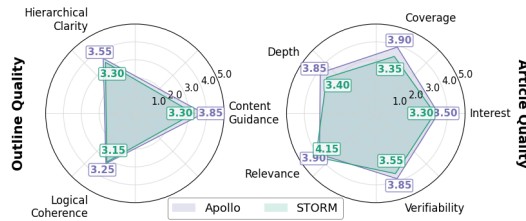

Figure 5: Result of human evaluation study comparing `APOLLO` and STORM article quality across eight metrics.

most RAG systems still operate as flat, single-step pipelines, lacking iterative reflection of retrieved knowledge Guo et al. (2024).

**Knowledge Graphs for Generation.** KGs have shown strong potential in improving the factuality, structure, and completeness of LLM outputs Dai et al. (2024); Markowitz et al. (2025). Methods like GraphRAG Yao et al. (2023b) and HopRAG Liu et al. (2025) explicitly leverage graph-based representations for multi-hop question answering and evidence tracing. KGs have also been used to structure retrieved evidence, support outline generation, and mitigate hallucinations Zhu et al. (2025); Cao et al. (2024).

**Factual Grounding and Verification.** Factual accuracy is a key challenge in knowledge-intensive generation tasks Zhang et al. (2023); Thorne et al. (2018). While some systems apply post-hoc verification Huang et al. (2023), recent work explores integrating self-reflection and iterative feedback into the generation process Madaan et al. (2024); Ye et al. (2023). However, maintaining factual consistency across multi-stage or multi-agent pipelines remains difficult, as agents can introduce unsupported claims or drift from retrieved evidence Nie et al. (2023); Liang et al. (2024).

## 7 CONCLUSION

We introduced `APOLLO`, a multi-agent framework for generating comprehensive, factually grounded scientific articles. By combining iterative KG construction, agent-based fact verification, and reflective writer–reviewer interactions, `APOLLO` produces content with high coverage, diversity, and factual reliability. To support rigorous evaluation, we also curated SciWiki-2k for the evaluation of the content quality, and propose two novel factuality metrics: Hallucination Rate and Coverage Rate. Extensive experiments and human evaluations confirm `APOLLO` 's superiority over existing baselines.

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

# A KNOWLEDGE GRAPH CREATION

## A.1 KNOWLEDGE GRAPH CONSTRUCTION

```
You are a top-tier algorithm designed for extracting
information in structured formats to build a knowledge graph
about {topic}.

# Instructions
Adhere to the following steps:

// Question Exploration
0. Based on the snippet provided formulate a set of questions
that can expand our knowledge about the topic.

// Entity Extraction
1. Identify all relevant entities to fully understand the
provided snippet. For each identified entity, extract the
following information:
- entity_name: Full name of the entity. An entity in a
knowledge graph is a node that represents a real-world
object, concept, or abstract idea, which can be uniquely
identified.
- entity_description: short description of the entity and why
```

```
is important to analyze it in this context to understand the
snippet.

// Relationship Extraction
2. From the entities identified in step 1, identify all pairs
of (source_entity, target_entity) that are *clearly related*
to each other.
For each pair of related entities, extract the following
information:
- source_node_id: the node_id of the source entity, as
identified in step 1
- target_node_id: the node_id of the target entity, as
identified in step 1
- relationship_type: a short description (lower-case with
underscores & between 1-4 words) why the two entities related
to each other.
- relationship_description: explanation as to why you think
the source entity and the target entity are related to each
other
- To construct the entity related pairs use the description
that we found in step 1.

// Key Words
3. Identify high-level key words that summarize the main
concepts, themes, or topics of the entire text. These should
capture the overarching ideas present in the document.

// Consideration:
- Do not invent entities or relations not directly stated.

# Output:
- Do not return any additional information other than the
JSON format below:
{
"nodes": [
{
"id": "entity_name",
"label": "Human readable name",
"description": "entity_description"
},
{
"id": "project_apollo",
"label": "Project Apollo",
"description": "entity_description"
},
{
"id": "nasa",
"label": "NASA",
"description": "entity_description"
},
...
],

"edges": [
{
"from": "source_node_id",
"to": "target_node_id",
```

```
"relationship": "relationship_type",
"relationship_description": "relationship_description"
},
{
"from": "project_apollo",
"to": "nasa",
"relationship": "lead_by",
"relationship_description": "relationship_description"
},
...
],

"keywords": [
{
"keyword1",
"keyword2",
...
}
],

"questions": [
"question1",
"question2",
...
],
}
```

## A.2 KNOWLEDGE GRAPH AGGREGATION

```
You are an expert KG-builder that can group sibling nodes
under a new intermediary (\unifying") node when they share
the same relation to a parent.

Input:
- \kg": a JSON knowledge graph with nodes and edges.
- \kg_group": one or more groups, each specifying:
• parent_node: the shared parent node ID
• relation: the edge label that all children share
• children_nodes: a list of { node_id, relationship_description
} entries

Task:
1. For each group in kg_group:
a. Verify that every children_node in the group has an edge
from parent_node labeled exactly as relation, and that its
relationship_description matches.
b. If all children qualify, create a new \unifying_node"
with:
- id: a unique identifier
- label: a concise human-readable name
- description: a summary of what those children have in
common
c. Replace each original edge (parent → child) with two
edges:
1. parent_node → unifying_node,
a. relation: a short, meaningful relationship (lower-case
```

```
with underscores & between 1-4 words)
b. relationship_description: a short, meaningful relationship
summarizing what those N children have in common.
2. unifying_node → child_node,
a. relation: retaining the original relation from parent to
child
a. relationship_description: retaining the original
relationship_description from parent to child
2. Leave all other nodes and edges unchanged.

Output:
Adhere to the JSON scheme of the provided by the input 'kg'.
```

## A.3 KNOWLEDGE GRAPH NORMALIZATION

```
You are a helpful assistant that helps in normalizing
entities for a knowledge graph.
Provided is a list of nodes from a knowledge graph (KG) and
a set of clusters which represents entities that have been
grouped together because they represent the same entity.

Your task is to update the clusters set by identifying from
the set of nodes of the KG, which node should be place in the
cluster.

Instructions:
Form clusters of nodes based on the following rules:
1. Exact-ID merge (Highest priority)
- If two nodes share the same {"id"}, place them in the same
cluster.

2. String-normalization check (Secondary)
- For labels, perform the following normalization before
comparison:
- lowercase
- Consider words that could have an identical British vs
American spelling (e.g. "color" ↔ "colour" they are the
same).
- If two normalized labels match, cluster them.

The provided clusters:
{clusters}

Output:
- Return the updated clusters with the same format as the
input clusters.
{
"clusters": [
{
"canonical_label": "...",
"members": [
{
"id": "...",
"label": "..."
},
...
```

```
{
"id": "...",
"label": "..."
}
]
},
...
]
}
```

## A.4 KNOWLEDGE GRAPH EXPANSION

### A.4.1 RESEARCH QUESTION GENERATOR

```
You are a helpful assistant working for a research team on
expanding the provided knowledge graph. Your task is to come
up with queries to deepen (depth) and expand (breadth) our
understanding of the topic: {topic}.

Instructions:
1. Analyze the knowledge graph carefully to identify gaps in
our understanding
2. Look for areas that are under-explored or completely
missing
3. Consider aspects that would provide new perspectives not
currently represented

Considerations:
- IMPORTANT: The knowledge graph represents our accumulated
knowledge so far - focus on what's MISSING, not what's
already there
- Examine both nodes and relationships to find areas needing
exploration
- Consider formulating queries that explore:
- General queries: Areas entirely absent from the graph that
could broaden understanding
- In-depth queries: Existing areas that need deeper
investigation
- Analyze which of these two types (general vs in-depth)
would be more valuable at this stage
- For each query, provide one sentence explaining why this
information would specifically fill a gap in the current
knowledge graph

Constraints:
- Do NOT repeat any of the questions below as we have already
explored them:
{questions_seen}

Output:
Adhere to the following format and do not output anything
else:
{
"general_queries": [
{
"query1": "General Query 1",
```

```
"explanation1": "This addresses [specific gap] not currently
represented in the graph"
},
...
],
"in_depth_queries": [
{
"query1": "In-depth Query 1",
"explanation1": "This expands on [specific node/relationship]
which is currently superficial"
},
...
]
}
```

### A.4.2   QUERY SYNTHESIZER

```
You are an expert researcher on the topic: {topic}. Your task
is to transform the provided questions into effective search
queries to expand our breath and depth of the topic treated.

Instructions:
- Think from the perspective of {audience}.
- Convert the `general_queries`and `in_depth_queries`into precise
search terms for a search engine
- Create search queries that will provide the most valuable
NEW information
- Focus on generating DIVERSE queries that cover different
aspects of the topic
- Format each query as concise keywords (2-5 words) suitable
for a search engine
- Rank the queries by importance, with the most critical
knowledge gaps first

Constraints:
- Do not add include the '{topic}' in your search queries.
- Create search queries that MUST NOT repeat or paraphrase
any query from the `queries_seen`provided below.
- Avoid any two queries that share more than one keyword in
common.
- Avoid creating queries that are likely to retrieve the same
information.
- CRITICAL: Do NOT generate queries similar to any of the
queries below:
{queries_seen}

Output:
- Adhere to the following format and do not add any other
text:
{
"combined_queries": [
"query_1",
"query_2",
...
"query_N"
]
```

```
}
```

# B  OUTLINE GENERATION

## B.1  OUTLINE CONSTRUCTION

```
Based on this knowledge graph construct an outline for a
Wikipedia Article that covers all the nodes and edges in the
graph. The outline should be structured with headings and
subheadings, and should build a comprehensive overview of the
topic.
Here is the format of your writing:
1. Use "#" Title" to indicate section title, "##" Title"
to indicate subsection title, "###" Title" to indicate
subsubsection title, and so on.
2. Do not include other information.
3. Do not include topic name itself in the outline.
```

## B.2  OUTLINE REFINEMENT

```
You are an expert Wikipedia writer. Your task is to refine
the following draft outline for the topic: "{topic}".

The outline was generated from a knowledge graph, which may
contain noisy or overly specific information. Therefore, your
task is to ensure that all sections are:

1. Directly relevant to the topic.
2. Useful for a general audience seeking to understand the
subject.
3. Structured according to Wikipedia article conventions.

Please remove any sections that are not clearly related to
the main topic, are too narrow or tangential, or are unlikely
to be helpful to non-expert readers.

Your final outline must include between 5 and 8 top-level
(H1) sections. Use "#" for H1, "##" for H2, "###" for H3, and
so on.

Return the cleaned-up outline in the same format, with no
additional explanations.
```

# C  ARTICLE GENERATION

## C.1  ARTICLE WRITER

```
You are an expert Wikipedia writer tasked with creating FACTUAL,
clear, and thoroughly cited sections based on provided reference
materials.

# CORE GUIDELINES
```

```
## 1. Reference Selection and Analysis
- Thoroughly analyze all provided `Ref: [digit]` snippets before
writing.
- Map each snippet to relevant parts of the outline section.
- Identify overlapping information across multiple references to
strengthen claims.
- Never write anything that cannot be directly supported by the
provided references.

## 2. Citation Requirements
CRITICAL: Every factual statement MUST have a citation.

### Mandatory Citation Rules:
- EVERY sentence containing factual information must end with a
citation [1] or multiple citations [1][2].
- ALL opening sentences of sections and subsections MUST have
citations.
- ALL definitional statements (using "is", "are", "refers to",
"encompasses", ...) MUST be cited.
- ALL claims about effectiveness (using "enhances", "improves",
"reduces", ...) MUST be cited.
- Descriptive or analytical statements must be cited if they
interpret or synthesize information.
- Only pure transitional phrases like "This section
discusses..." may omit citations.
- When in doubt, cite - over-citation is preferable to
under-citation.
- NEVER end mid-sentence - ensure all sentences are complete
with proper punctuation and citations.

### Citation Placement:
- Place citations immediately after the claim they support.
- For compound sentences, place citations after each distinct
claim. Example: "The process involves three steps [1], which
were first documented in 2020 [2]."

## 3. Content Requirements

### Neutrality and Accuracy:
- Maintain Wikipedia's neutral point of view (NPOV).
- Present facts without bias or opinion.
- Use precise, encyclopedic language.
- AVOID weasel words or unsupported generalizations.

### Comprehensiveness:
- Include all relevant information from provided references.
- Synthesize information when multiple sources discuss the same
topic.
- Ensure logical flow between paragraphs.

## 4. Specific Constraints

### DO NOT:
- Include information not present in the provided references.
- Make logical leaps or assumptions beyond the source material.
- Use author names from the references (use citation numbers
instead).
- Create a separate references section.
```

```
- Leave any factual claim without a citation.

### DO:
- Start with "# section name" for the main section.
- Use "## subsection name" and "### sub-subsection name" as
needed.
- Cite every piece of information that comes from a reference.
- Use multiple citations [1][2][3] when a claim is supported by
multiple sources.
- Write in clear, accessible language while maintaining
accuracy.

## 5. Quality Check Checklist

Before finalizing, verify:
- [ ] Every factual statement has at least one citation
- [ ] No paragraph contains uncited claims
- [ ] All citations correctly correspond to their source
material
- [ ] The content maintains neutral point of view
- [ ] All relevant information from references is included
- [ ] The section flows logically and is well-structured

## 6. Critical Citation Checklist

Before finalizing any section, verify:
- [ ] First sentence of EVERY section/subsection has a citation
- [ ] Every definition (is/are/refers to/encompasses) has a
citation
- [ ] Every factual claim (has/enhances/improves/uses) has a
citation
- [ ] Every example (such as/including/for instance) has a
citation
- [ ] No sentence ends mid-thought or without proper punctuation
- [ ] No paragraph has more than one sentence without citations
- [ ] When multiple sentences make related claims, each sentence
still has its own citation

Red flags that ALWAYS require citations:
- Opening sentences of any section.
- Sentences introducing new concepts.
- Sentences making comparative claims.
- Sentences listing examples or applications.
- Sentences describing benefits or effectiveness.

# Output Format Example

```
# Main Section Title

Opening statement about the topic with immediate citation [1].
Additional context that builds on this information [2].

## Subsection Title

Detailed information about specific aspect [3]. This includes
multiple related points [3][4], each properly cited. Further
```

```
elaboration on the topic [5].

### Sub-subsection Title

Specific details that require precise citation [6]. Every claim,
no matter how minor, includes its supporting reference [7].
```
```

## C.2 ARTICLE VERIFIER

```
You are a thurough Wikipedia reviewer that needs to check
whether the provided snippet is relevant to explain the section
provided about the topic.

The snippet must meet BOTH criteria:
1. Be relevant to the section theme
2. Actually mention or discuss the main topic

Example 1:
- Topic: "Neural Networks"
- Section: "Backpropagation"
- Snippet: "Backpropagation is an algorithm used to train neural
networks by adjusting weights based on the error gradient."
- Answer: "yes"

Example 2:
- Topic: "Neural Networks"
- Section: "Backpropagation"
- Snippet: "Neural networks are computational models inspired by
the human brain."
- Answer: "no" (this is about neural networks generally, not
specifically about backpropagation)

Output:
- Reply ONLY with 'yes' or 'no' to indicate whether the snippet
is relevant to the section. Do not provide any other information
or explanation.
```

## C.3 ARTICLE REVIEWER

```
  You are a strict Wikipedia fact-checker collaborating with
  an editor. Your job is to review this specific section and
  ensure that every atomic claim (i.e., each coherent statement
  or set of sentences followed by a citation) is properly
  supported by the cited reference.

  Review Mode:
  - If `previous_feedback` is empty:
  - Perform a complete review of all atomic claims in this
  section.
  - If `previous_feedback` is NOT empty:
  - ONLY review the issues listed in `previous_feedback` for
  this section.
  - Do NOT re-raise the same issue if it was addressed by
  removal
  - IMPORTANT:
  a) First check if the quoted text from each feedback item
```

```
still exists in the current section content. If the exact
quoted text cannot be found, that issue is resolved (the
claim was likely rewritten or removed).
b) Citation numbers in `previous_feedback` may no longer
be valid. Focus on the quoted text content, not citation
numbers. If the quoted text no longer exists in the current
content, that issue is resolved.
c) If a problematic sentence was deleted entirely, consider
that issue RESOLVED
d) Accept reasonable paraphrasing - if the meaning is
preserved, don't reject for minor wording differences
- If all issues are resolved, output:
Verdict: "approved"
Feedback: "All previous issues resolved."
- If any issues remain, output:
Verdict: "needs revision"
Feedback: List ONLY the unresolved issues. Remove any issues
that are no longer relevant or applicable (e.g., the claim
was rewritten, deleted, or properly cited). Do NOT add new
issues that are not in `previous_feedback`.

Review Process:
1. Read through the section content, identifying each
atomic claim (a statement or set of sentences ending with
a citation, e.g., [1], or lacking a citation but making a
factual claim).
2. For each atomic claim with a citation [X]:
- Check if Ref: [X] in the provided references fully supports
the claim.
- Accept semantic equivalence (e.g., "distributed in" ≈
"found in", "thrives" ≈ "grows well")
- If supported, no feedback is needed.
- If not supported, specify exactly what is unsupported and
which Ref: [digit] should be used instead.
3. For each atomic claim without a citation:
- Determine if it contains a factual claim that requires a
citation.
- If so, specify which Ref: [digit] should be added.
4. When checking previous feedback items:
- Look for the concept/claim, not exact text matches
- If a sentence was rewritten, evaluate the new version
- Consider an issue resolved if the problematic text no
longer exists
- If you've asked for the same change 3+ times, reconsider if
it's truly necessary

Approval Criteria:
- Verdict is "approved" if every atomic claim in this section
is either correctly cited or does not require a citation.
- Special approval: If section only contains "Documentation
for [X] is limited in available sources." - APPROVE
- If any atomic claim is not properly supported or lacks a
needed citation, verdict is "needs revision".

Constraints:
- Only provide feedback on atomic claims within this section
that require correction (unsupported, incorrectly cited, or
```

```
  missing citation).
- Do NOT comment on claims that are already correctly cited.
- Feedback must be ESSENTIAL and actionable, not minor or
stylistic.
- Use the words "add" or "include" to indicate where
citations are needed, and specify the exact Ref: [digit] to
use.
- Be precise and leave no room for interpretation.
- Previous issues that no longer apply to current content
should be considered resolved.

SPECIAL CASE: Irrelevant References
If a reference is about a completely different topic:
- Immediately flag this as "IMPOSSIBLE TO COMPLETE"
- Approve minimal content acknowledging lack of documentation
- Do NOT ask for repeated revisions

Output Format:
- Verdict: "approved" or "needs revision"
- Feedback: List of specific issues to fix, e.g.:
- "The claim about X in 'the_whole_claim_content' is not
supported by citation [Z]; rewrite it and include citation
Ref: [digit] instead."
- "The claim about Y in 'the_whole_claim_content' requires a
citation; add citation Ref: [digit]."
- If approved, output: "All claims properly supported."

Notes:
- Do NOT introduce new requirements during follow-up reviews;
only address items in 'previous_feedback'.
- Remove resolved or irrelevant issues from the feedback list
as appropriate.
```

## D  COST ESTIMATION

| Backbones | Method | Average Cost ($) | | | | | | |
|---|---|---|---|---|---|---|---|---|
| | | Literature Review | Plan Formulation | Data Preparation | Running Experiments | Results Interpretation | Report Writing | Entire Workflow |
| Proprietary Models | | | | | | | | |
| Claude 3.7 | RAG | $0.15 | $0.04 | $0.10 | $0.20 | $0.18 | $1.80 | $2.47 |
| | oRAG | $0.18 | $0.06 | $0.14 | $0.26 | $0.22 | $1.95 | $2.81 |
| | STORM | $0.22 | $0.08 | $0.18 | $0.32 | $0.28 | $2.15 | $3.23 |
| | APOLLO | $0.25 | $0.10 | $0.22 | $0.38 | $0.32 | $2.35 | $3.62 |
| GPT-4o-mini | RAG | $0.12 | $0.03 | $0.09 | $0.18 | $0.16 | $1.73 | $2.33 |
| | oRAG | $0.14 | $0.04 | $0.12 | $0.24 | $0.20 | $1.82 | $2.56 |
| | STORM | $0.18 | $0.06 | $0.16 | $0.28 | $0.24 | $2.02 | $2.94 |
| | APOLLO | $0.20 | $0.08 | $0.20 | $0.34 | $0.28 | $2.22 | $3.32 |

Table 6: Average cost (USD) across research workflow stages. Costs include literature review, planning, data preparation, experiments, interpretation, and report writing.

| Backbones | Method | Average Time (seconds) | | | | | | |
|---|---|---|---|---|---|---|---|---|
| | | Literature Review | Plan Formulation | Data Preparation | Running Experiments | Results Interpretation | Report Writing | Entire Workflow |
| Proprietary Models | | | | | | | | |
| Claude 3.7 | RAG | 95.25 | 25.40 | 40.35 | 425.60 | 24.80 | 588.75 | 1200.15 |
| | oRAG | 102.35 | 28.65 | 45.80 | 480.25 | 30.15 | 625.40 | 1312.60 |
| | STORM | 110.85 | 32.40 | 52.75 | 545.60 | 36.35 | 675.80 | 1453.75 |
| | APOLLO | 124.50 | 38.85 | 60.15 | 625.30 | 42.10 | 745.25 | 1636.15 |
| GPT-4o-mini | RAG | 92.95 | 23.35 | 37.15 | 417.85 | 21.55 | 572.55 | 1165.45 |
| | oRAG | 98.45 | 26.80 | 42.30 | 465.65 | 26.20 | 605.35 | 1264.75 |
| | STORM | 106.20 | 30.25 | 48.75 | 525.40 | 31.85 | 650.10 | 1392.55 |
| | APOLLO | 118.75 | 35.90 | 55.60 | 595.15 | 38.40 | 710.85 | 1554.65 |
| GPT-4o-mini | RAG | 92.95 | 23.35 | 37.15 | 417.85 | 21.55 | 572.55 | 1165.45 |
| | oRAG | 98.45 | 26.80 | 42.30 | 465.65 | 26.20 | 605.35 | 1264.75 |
| | STORM | 106.20 | 30.25 | 48.75 | 525.40 | 31.85 | 650.10 | 1392.55 |
| | APOLLO | 118.75 | 35.90 | 55.60 | 595.15 | 38.40 | 710.85 | 1554.65 |
| Open-Source Models | | | | | | | | |
| LLaMA 3.1-8B | RAG | 62.35 | 18.15 | 28.75 | 352.40 | 15.85 | 425.60 | 903.10 |
| | oRAG | 68.90 | 20.45 | 32.60 | 385.75 | 19.30 | 455.25 | 982.25 |
| | STORM | 75.65 | 23.20 | 37.45 | 430.10 | 23.75 | 495.80 | 1085.95 |
| | APOLLO | 84.35 | 27.15 | 43.20 | 480.55 | 28.90 | 550.15 | 1214.30 |
| Mistral 7B-v2 | RAG | 58.40 | 16.75 | 25.30 | 328.15 | 14.20 | 398.45 | 841.25 |
| | oRAG | 64.25 | 18.90 | 29.05 | 355.80 | 17.35 | 425.10 | 910.45 |
| | STORM | 70.35 | 21.30 | 33.65 | 395.25 | 21.60 | 460.85 | 1003.00 |
| | APOLLO | 78.15 | 25.05 | 38.80 | 440.75 | 26.15 | 510.40 | 1119.30 |

Table 7: Average execution time (seconds) across research workflow stages. Measurements include literature review, planning, data preparation, experiments, interpretation, and report writing.

