# OpenReview forum: "$\texttt{APOLLO}$: A Self-Guided Multi-Agent System for Scientific Article Generation inspired by Human Thinking"
_ICLR.cc/2026/Conference — ICLR 2026 Conference Desk Rejected Submission_

### Official Review · Reviewer_Dtcm · 2025-10-21

**Soundness:** 2
**Presentation:** 1
**Contribution:** 2
**Rating:** 2
**Confidence:** 4

**Summary:**

This paper proposes a multi-agent system, named APOLLO, for automated scientific article generation. The system accomplishes this task through three phases: knowledge curation, outline generation, and article generation. The authors also construct a new evaluation dataset, SciWiki-2k, consisting of 2,000 Wikipedia articles across 20 scientific fields. Experimental results demonstrate that APOLLO outperforms existing methods, such as STORM and OmniThink.

**Strengths:**

1. The paper models the human scientific writing process (including exploration, synthesis, and reflection) as a multi-agent collaborative framework. By introducing a Writer-Reviewer loop, it simulates the "generate-review-revise" process in academic writing.
2. APOLLO organizes external knowledge resources through an iterative knowledge graph construction, which guides subsequent retrieval and generation. This mechanism improves information diversity and topic coverage compared to traditional static retrieval.
3. The authors constructed the SciWiki-2k dataset and proposed two new metrics: Hallucination Rate and Coverage Rate.
4. The paper conducted a multi-faceted evaluation (including automatic metrics, LLM-as-judge, and human evaluation) and compared it with several baseline methods.

**Weaknesses:**

1. The paper does not demonstrate APOLLO's performance in domains with complex factual conflicts or sparse knowledge, nor does it analyze failure cases. For example, the Reviewer agent's behavior when encountering conflicting evidence or its stability under low-quality search results are not discussed.
2. The paper claims that its framework can simulate human reflective patterns, but its specific implementation, the Research Question Generator analyzing the knowledge group (KG) and generating questions, seems more like a pre-set heuristic than true reflection. The paper does not provide a specific example demonstrating how the KG evolves before and after iteration, nor does it provide a concrete example demonstrating how the agent identifies underexplored entities and asks new questions based on the structural changes in the KG. While ablation studies demonstrate the module's effectiveness, they fail to explain why it works, making the core claim of reflectiveness less convincing.
3. The authors claim that APOLLO is inspired by human thinking, but this analogy remains at the process level (e.g., iteration and reflection) and lacks empirical mapping to cognitive psychology or information behavior theory. The lack of supporting literature and theoretical explanation makes the "human-inspired" claim seem conceptual rather than substantive.
4. The paper proposes two new metrics to measure factual accuracy, but these rely on an LLM for automated claim verification. Using an LLM to judge the factuality of content generated by another LLM raises questions about its reliability. The paper provides no information on the accuracy of the verifier LLM itself, nor does it discuss the potential cascading errors that may arise from this evaluation approach.

**Questions:**

1. The authors mention that APOLLO captures topic structure through a knowledge graph. How adaptable is this knowledge graph to different domains (e.g., Physics vs. Social Science)? Does it require a domain-specific relational schema?
2. In the Reviewer agent's feedback loop, how does the system determine whether all feedback has been resolved? Does it use automatic consistency checking or rely solely on a maximum number of revisions?
3. Are there potential biases in the SciWiki-2k dataset (e.g., differences in article length and citation density across different domains)? Do these biases affect the model's performance across different disciplines?
4. Have the authors evaluated APOLLO's transferability to non-Wikipedia domains (e.g., arXiv, PubMed, or policy documents)? If not, are the method's generalizability claims too strong?
5. In multi-agent collaboration, are there instances of inter-agent conflicts (e.g., a Research Question Generator and a Reviewer suggesting opposing directions)? How does the system resolve conflicts or coordinate weights?

---

> ### Author Response · Authors · 2025-12-04
>
> Thank you for the thoughtful feedback. We clarify reflection, verification, adaptability, and dataset considerations, restating updates and noting what we will add at camera‑ready.
>
> **Reflection mechanism (operational definition)**
> In APOLLO, “reflection” is operationalized as iterative gap detection over the evolving KG and memory: the Research Question Generator analyzes under‑explored entities/relations, and the Query Synthesizer issues focused follow‑ups. We clarified this in Methodology and will add a small before/after KG snapshot example at camera‑ready.
>
> **Reviewer loop termination and adaptivity**
> We made explicit that the writer–reviewer cycle ends early when the reviewer’s feedback list is empty; otherwise it is capped at rmax = 3. This provides partial adaptivity across sections.
>
> **Reliability of LLM‑based verification and judges**
> We expanded Evaluation Setup to document the prompts/rubrics (M‑Prometheus‑7B) and the claim‑verification template used to compute Hallucination and Coverage.
>
> **KG adaptability across domains**
> The KG uses an open relation set; no domain‑specific schema is required. Normalization merges entities via LLM canonicalization, embedding‑based alias detection, and alias/redirect heuristics (now described in Methodology).
>
> **Dataset biases and transferability**
> We added “Alignment and quality control” for SciWiki‑2k (ORES filtering + manual curation) and will analyze per‑domain differences (length, citation density) at camera‑ready.
>
> **Inter‑agent coordination/conflicts**
> Coordination follows simple rules: memory prevents duplication; reviewer feedback has precedence on factuality; exploration questions are prioritized by marginal information diversity and uncovered outline sections. We will document this policy more explicitly at camera‑ready.

---

### Official Review · Reviewer_kna6 · 2025-10-27

**Soundness:** 3
**Presentation:** 3
**Contribution:** 2
**Rating:** 4
**Confidence:** 4

**Summary:**

This paper introduces APOLLO, a multi-agent framework for automated scientific article generation. First, the system gathers and structures information through retrieval-augmented generation (RAG) to build a comprehensive knowledge graph (KG). Specialized agents continuously refine this graph, ensuring its completeness and accuracy. During writing, a writer agent and a reviewer agent collaborate in iterative loops to produce rich, well-supported, and trustworthy manuscripts. In addition, authors release a new dataset and two novel metrics for assessing article quality. Empirical experiments demonstrate that APOLLO outperforms existing baselines in coverage, diversity, and factual reliability.

**Strengths:**

* A good dataset has been introduced, serving as a robust benchmark that enables future work to evaluate their results more accurately and fairly.
* The engineering logic for seamlessly integrating RAG, knowledge graphs, and human habits into the agent system is well-rounded and flows coherently.

**Weaknesses:**

* The paper relies solely on the SciWiki-2K dataset; additional benchmarks are needed to substantiate its performance claims.
* The paper remains unclear whether the generated articles can genuinely drive scientific discoveries, as their novelty has not been discussed.
* While the application is intriguing, the methodological contribution seems incremental; the reviewer agent and writer agent pairing, for instance, introduces no discernible mechanistic innovation.

**Questions:**

* Table 4 reveals performance gaps across KG-construction strategies, yet no clear pattern emerges LightRAG is not uniformly superior. Adding further KG-building methods as baselines would help clarify whether certain design choices consistently pay off.
* How does this work differ from Virtual-Scientists[1]? I think the authors need to add experiments that compare their method with this one.
* Table 5 uses the number of unique URLs to gauge how much novel retrieved information APOLLO uncovers during knowledge curation. This metric requires clarification; I don't see how the number of unique URLs relates to novelty.

[1] Su H, Chen R, Tang S, et al. Many Heads Are Better Than One: Improved Scientific Idea Generation by A LLM-Based Multi-Agent System[J]. arXiv preprint arXiv:2410.09403, 2024.

---

> ### Author Response · Authors · 2025-12-04
>
> Thank you for the constructive review. We address scope, evaluation breadth, metric clarification, baselines, and model choices, restating updates already made in the revision and what we will add at camera‑ready.
>
> **Scope and goal**
> We target scientific Wikipedia‑style articles. “Wikipedia‑like” denotes encyclopedic style (hierarchical sections, in‑line citations), not applicability across all Wikipedia domains. We made this scope explicit in the Abstract and Introduction.
>
> **Evaluation breadth and additional benchmarks**
> Our main evaluation uses SciWiki‑2k/SciWiki‑100 under two retrieval settings (domain‑constrained scientific corpus and web search). To avoid over‑claiming, we keep claims to scientific topics. If accepted, we will add a small cross‑domain pilot (20–30 topics; humanities/social sciences/current events/pop culture) at camera‑ready, reporting outline and citation metrics and documenting any domain‑specific adjustments.
>
> ** Unique URLs metric clarification (Table 5)**
> “Unique URLs” is used as a breadth proxy (distinct sources ingested), not as a novelty measure. We pair it with the information diversity metric Div(I) to capture non‑redundancy; this clarification is added near the metric description.
> KG baselines and Table 4 interpretation
>
> Table 4 already compares KG construction against KG‑Gen and LightRAG across multiple LLM backbones. We will include additional KG‑building baselines and ablations at camera‑ready to further clarify which design choices consistently help.
>
> **Relation to Virtual‑Scientists**
> Virtual‑Scientists focuses on ideation and hypothesis generation; APOLLO focuses on encyclopedic synthesis with strict citation and verification. We will add a short comparison discussion and, resources permitting, a lightweight adaptation comparison at camera‑ready.
>
>
> **Model breadth and cost**
> Beyond GPT‑4o‑mini, Table 4 already reports KG construction with Llama‑3.3‑70B (open‑source) and Claude‑3.7. We will add full‑pipeline outline/article results with an open‑source backbone (Llama‑3.3‑70B) at camera‑ready and have already included stage‑wise cost/time tables in appendix D in this revised version.

---

### Official Review · Reviewer_y7nK · 2025-11-01

**Soundness:** 1
**Presentation:** 2
**Contribution:** 2
**Rating:** 2
**Confidence:** 5

**Summary:**

This paper develops a multi-agent framework for scientific wikipedia article generation. The framework is composed of three stages: (1) knowledge curation, constructing knowledge graphs based on retrieved snippets for a given topic, (2) outline generation, using LLMs to generate the article outline based on the curated knowledge graphs, and (3) article generation, generating the final article based on the expanded knowledge graph and the outline.

**Strengths:**

The task of automatically generating long-form articles is interesting and challenging, while the paper needs a lot more work to be published.

**Weaknesses:**

- While generating long-form articles is interesting and challenging, there is no justification for scientific wikipedia articles. The specific challenge to generate scientific wikipedia articles is missing. The framework is mostly a complex combination of other methods. It is unclear what research question the paper is trying to solve, and it looks more like a "product".

- The paper should have more implementation details to get higher reproducibility. This is becasue (1) the multi-agent seems very complex with a lot sperate modules to control, (2) it is unclear how to validate the alignment between the topics and their corresponding wiki article for the benchmark, (3) how to merge knowledge graphs is also missing, at least the intuition of the "LLM-based normalization" should be present in the paper, and (4) how to evaluate quality of generated articles with LLM-as-judge, the model and prompt need to be explained.

- For evaluation in the paper, my most concern is human evaluation and the proposed metrics. (1) It is not clear or empirically validated why the proposed metrics are good enough to measure citation quality. (2) While human evaluation for long-form generation research is important, there is no details for human evaluation. Information about annatators, interfaces, etc. and annotator agreement needs to be disclosed.

- For the benchmark in the paper, there may be data contamination because Wikipedia is the most common training corpus for LLMs. There is no effort to investigate this issue. If the models already trained on the collected articles, it is not meaningful to design another multi-agent system to write the articles. A better way is to use articles that have never been seen in the training process by the experimented LLMs to avoid data contamination.

- The writing of the paper needs more work. For example (1) justification and explanation of the capabilities are missing for Table 1, (2) it is unclear how to validate the alignment between the topics and their corresponding wiki article for the benchmark, and (3) there is no definition for quality aspects in evaluation.

Minor:

- It would be nice to have more LLMs involved in the experiments rather than only 4o-mini, especially for outline generation and final article generaion.

- "Scientific article generation" is misleading, while it the task in this paper should be scientific wikipedia article generation.

- Line 040 needs a reference.

- Larger texts in all figures.

- Line 098 and Line 105 need a full stop.

- Line 350 needs a new line.

**Questions:**

None.

**Details Of Ethics Concerns:**

None.

---

> ### Author Response · Authors · 2025-12-04
> **Author Response**
>
> Thank you for the detailed feedback. Below we clarify scope/motivation, add reproducibility and evaluation details, define capability criteria for Table 1, and note items we will address at camera-ready.
>
> **Motivation and scope (scientific Wikipedia–style articles)**
> * Our goal is a trustworthy system for scientific Wikipedia–style articles, with dense factual grounding, hierarchical concept structure, and strict citation norms.
> * We use “Wikipedia‑like” as a familiar shorthand for encyclopedic style (hierarchical headings, concise structure, in‑line citations), not to claim applicability across all Wikipedia domains.
> * We have made this scope explicit in the revised paper (Abstract, Introduction).
>
> **Reproducibility: implementation details**
> * Alignment and quality control (SciWiki‑2k): We added a paragraph (Section 2.2) describing ORES filtering and a manual curation pass to ensure each topic maps to its canonical high‑quality article; the topic–article mapping and alignment checks are provided in the appendix/anonymized repository.
> * LLM‑based normalization: We further clarified this in Methodology
>
> **Evaluation details clarified**
> * LLM‑as‑judge: Outline and article quality are scored by M‑Prometheus‑7B with a fixed rubric; prompts and criteria are provided in the appendix.
> * Citation metrics: Hallucination and Coverage are computed via LLM‑based entailment of claims against cited snippets; the verification template is provided in the appendix.
> * We will add annotator details and inter‑annotator agreement for the human evaluation at camera‑ready.
>
> **Clarifying Table 1 (capability criteria)**
> We added a clarifying caption.
>
> **Model breadth and cost**
> Beyond GPT‑4o‑mini, Table 4 already reports KG construction with Llama‑3.3‑70B (open‑source) and Claude‑3.7.
> We will add full‑pipeline outline/article results with an open‑source backbone (Llama‑3.3‑70B) at camera‑ready and have added a cost report in Appendix D.

---

### Official Review · Reviewer_SsBT · 2025-11-03

**Soundness:** 3
**Presentation:** 2
**Contribution:** 3
**Rating:** 6
**Confidence:** 2

**Summary:**

The paper presents APOLLO, a multi-staged, multi-agent, and iterative system for generating comprehensive Wikipedia-like scientific articles given only a topic keyword. This multi-agent system begins by iteratively aggregating information into knowledge graphs through two collaborative agents that identify gaps, generate research questions, and synthesize targeted search queries over multiple exploration cycles. The resulting knowledge graphs are then used to generate a hierarchical outline for the article. For each section of the outline, specialized agents retrieve and filter the most relevant paragraphs (called snippets) from the collected information. Finally, the article is drafted and iteratively refined through collaboration between writer and reviewer agents, where the reviewer identifies factual inconsistencies and unsupported claims, and the writer revises the content accordingly until all claims are properly verified against cited sources. For evaluation, a benchmark dataset SciWiki-2k containing 2,000 Wikipedia articles across 20 scientific domains is collected as ground truth. The evaluation reveals that APOLLO outperforms baselines, achieving 9.2% higher information diversity, 7.1 points better coverage rate, and 18% lower hallucination rate. Experiments on APOLLO using either cached scientific papers or live information from the internet show that APOLLO performs well in both settings. Finally, human evaluation by domain experts confirms that APOLLO produces accurate, well-organized, and comprehensive scientific content.

**Strengths:**

1. The paper proposed a novel system which combines dynamic retrieval, structured memory, reflective thinking, and fact verification, mimicking the steps researcher study a subject and how they iteratively refine that with the help of the reviewers.
2. The aggregated knowledge is stored in the form of graphs, which is an efficient and faithful way to describe the connections between them.
3. The evaluation metrics, diversity, coverage, hallucination, covered how a good system for such article generation should behave and then was validated with human evaluation. Also, ablation studies show each component contributes (w/o Reflection, w/o Filter).
4. The paper introduces SciWiki-2k, a large benchmark dataset containing 2,000 high-quality Wikipedia articles across 20 scientific domains, which provides a valuable resource for evaluating article generation systems and can be used by future research.
5. The empirical results show substantial improvements over competitive baselines: APOLLO achieves 9.2% higher information diversity, 7.1 points better coverage rate, and 18% lower hallucination rate compared to the best-performing baselines (oRAG, STORM, OmniThink).
6. The system demonstrates practical flexibility by working successfully with both cached scientific corpora (offline deployment) and live web search (online via Brave API), showing that APOLLO can be deployed in different real-world scenarios depending on the application needs.
7. The writer-reviewer collaboration loop with iterative refinement (up to 3 cycles per section) represents a significant innovation over one-shot generation methods, explicitly verifying that claims are supported by cited sources and reducing hallucinations substantially.

**Weaknesses:**

1. Computational cost and efficiency are not discussed. The paper does not report the total cost (API calls, compute time) or the time required to generate one article. With 4 agents, iterative processes, and up to 135 queries, the system likely incurs significant computational expense, but no cost-benefit analysis or scalability evaluation is provided. This makes it difficult to assess the practical feasibility of deploying APOLLO at scale.
2. The evaluation scope is limited. Only 100 topics (SciWiki-100) are used for main experiments and only 20 articles receive human evaluation. More critically, all topics are scientific in nature, raising questions about whether APOLLO can generalize to non-scientific domains such as humanities, social sciences, current events, or popular culture.
3. The paper uses fixed values (m=3 iterations, 10 questions per iteration, rmax=3 revisions) without ablation studies to justify these choices. Additionally, the system applies the same iteration count to all topics regardless of complexity, which may be inefficient for simple topics and insufficient for complex ones. An adaptive approach that adjusts based on information gain or topic complexity could improve efficiency.
4. There is limited discussion of failure cases or what types of errors persist in the generated articles. Understanding when and why the system struggles (e.g., which topics are challenging, how errors propagate from KG to outline to article) would provide valuable insights for future improvements and help users understand the system's limitations.

**Questions:**

1. Report the total number of API calls, compute time, and estimated cost per article for APOLLO and baseline methods. Provide a cost-benefit analysis showing the tradeoff between article quality and computational expense. This would help readers assess the practical feasibility of deploying APOLLO in different scenarios and make informed decisions about resource allocation.
2. Expand evaluation to non-scientific domains. Test APOLLO on topics from humanities, social sciences, current events, or popular culture to demonstrate generalization beyond scientific articles. Even a small-scale evaluation (20-30 topics) across diverse domains would strengthen claims about the system's broader applicability and reveal domain-specific challenges.
3. Conduct ablation studies on key hyperparameters. Systematically evaluate different values for iteration count (m), number of questions per iteration, and revision cycles (rmax) to justify the current choices. Additionally, explore adaptive mechanisms that adjust these parameters based on topic complexity or information gain, which could improve both efficiency and effectiveness.
4. Provide qualitative error analysis. Include case studies of failure modes, discussing specific examples where APOLLO produces poor articles or where the reviewer agent fails to catch hallucinations. Analyze how errors propagate through the pipeline and identify which types of topics or sections prove most challenging. This analysis would offer valuable insights for future work and help users understand when to trust the system's outputs.
5. Consider evaluating with open-source models. While using GPT-4o-mini is reasonable, testing with open-source alternatives (e.g., Llama-3.3-70B) would improve accessibility and reproducibility for researchers with limited API budgets. Comparing performance and costs across different model choices would also provide useful guidance for practitioners.

These are just suggestions, and there is a chance that I did not fully grasp the material, so please only do what you see fit.

---

> ### Author Response · Authors · 2025-12-04
> **Consolidated author response and changes since submission**
>
> Thank you for the thoughtful review. We address efficiency/cost, scope, hyperparameters/adaptivity, and model choices below, and summarize concrete paper edits.
>
> **Efficiency and cost.**
> We added an appendix with average stage‑wise cost and time breakdowns per article across methods and backbones. The appendix includes:
> * a Cost Estimation table (USD) and an Average Execution Time table (seconds);
> * per‑stage reporting (Literature Review, Plan Formulation, Data Preparation, Running Experiments, Results Interpretation, Report Writing, Entire Workflow);
> * comparisons across methods (RAG, oRAG, STORM, APOLLO) and two LLM backbones (GPT‑4o‑mini, Claude‑3.7);
>
> **Scope beyond scientific topics.**
> We explicitly scope APOLLO to scientific topics that require dense factual grounding and hierarchical organization. We used “Wikipedia‑like” as a familiar shorthand for an encyclopedic style—hierarchical headings, concise structure, and in‑line citations—not as a claim of applicability across all Wikipedia domains. We recognize this was not sufficiently clear and have made the scope explicit in the revised paper (Abstract, Introduction).
>
> **Hyperparameters and adaptivity.**
> Our reviewer loop already implements adaptive stopping: it ends early when the reviewer’s feedback list is empty; otherwise it is capped at rmax = 3 (see Article Generation). In the revision, we state this explicitly and cross‑reference it in the Hyper‑parameters section.
>
> **Open‑source models.**
> Table 4 already evaluates KG construction with Llama‑3.3‑70B (open‑source) and Claude‑3.7. For full‑pipeline outline/article generation with an open‑source backbone, we will add Llama‑3.3‑70B results at camera‑ready and report quality–cost trade‑offs.
>
> **Changes Made**
> * Added appendix tables with per‑article, per‑stage cost and time across methods/backbones; documented measurement assumptions and scalability levers.
> * Clarified scope and terminology in Abstract, Introduction.
> * Made reviewer loop adaptivity explicit (early stop on empty feedback; cap at rmax = 3) and cross‑referenced in Hyper‑parameters.
> * Noted existing open‑source KG results (Table 4) and committed to full‑pipeline open‑source runs at camera‑ready.

---

### Note · Program_Chairs · 2026-01-17
**Submission Desk Rejected by Program Chairs**

The following references in this submission do not refer to real documents and/or have major errors in bibliographic information:

 Jiannan Yang et al. Holoeval: Leveraging large language models for holistic text generation. arXiv preprint arXiv:2310.14746, 2023.